# Effect of Local Topography on Cell Division of *Staphylococcus* spp.

**DOI:** 10.3390/nano12040683

**Published:** 2022-02-18

**Authors:** Ioritz Sorzabal-Bellido, Luca Barbieri, Alison J. Beckett, Ian A. Prior, Arturo Susarrey-Arce, Roald M. Tiggelaar, Joanne Fothergill, Rasmita Raval, Yuri A. Diaz Fernandez

**Affiliations:** 1Surface Science Research Centre and Open Innovation Hub for Antimicrobial Surfaces, Department of Chemistry, University of Liverpool, Liverpool L69 7ZD, UK; isorzabal@unav.es (I.S.-B.); luca.barbieri@liverpool.ac.uk (L.B.); 2Institute of Infection and Global Health, University of Liverpool, Liverpool L69 3BX, UK; j.fothergill@liverpool.ac.uk; 3Biomedical Electron Microscopy Unit, University of Liverpool, Liverpool L69 3BX, UK; alib@liverpool.ac.uk (A.J.B.); iprior@liverpool.ac.uk (I.A.P.); 4Mesoscale Chemical Systems, MESA+ Institute, University of Twente, 7522 NB Enschede, The Netherlands; a.susarreyarce@utwente.nl; 5NanoLab Cleanroom, MESA+ Institute, University of Twente, 7522 NB Enschede, The Netherlands; r.m.tiggelaar@utwente.nl

**Keywords:** surface topography, bacterial cell growth mode, *Staphylococcus* spp., vertically aligned silicon nanowire arrays, early stage biofilm

## Abstract

Surface engineering is a promising strategy to limit or prevent the formation of biofilms. The use of topographic cues to influence early stages of biofilm formationn has been explored, yet many fundamental questions remain unanswered. In this work, we develop a topological model supported by direct experimental evidence, which is able to explain the effect of local topography on the fate of bacterial micro-colonies of *Staphylococcus* spp. We demonstrate how topological memory at the single-cell level, characteristic of this genus of Gram-positive bacteria, can be exploited to influence the architecture of micro-colonies and the average number of surface anchoring points over nano-patterned surfaces, formed by vertically aligned silicon nanowire arrays that can be reliably produced on a commercial scale, providing an excellent platform to investigate the effect of topography on the early stages of *Staphylococcus* spp. colonisation. The surfaces are not intrinsically antimicrobial, yet they delivered a topography-based bacteriostatic effect and a significant disruption of the local morphology of micro-colonies at the surface. The insights from this work could open new avenues towards designed technologies for biofilm engineering and prevention, based on surface topography.

## 1. Introduction

Bacteria display a variety of shapes and sizes determined by different mechanisms guiding their cell growth and division [1]. The cell shape of most bacterial species is maintained by a cell wall formed mainly by peptidoglycan molecules, providing stability against differential osmotic pressure, preventing cell lysis, and also allowing changes in cell shape and size during growth and division [2]. The location of proteins within the peptidoglycan cell wall is highly regulated in terms of spatial organisation during the cell division process. This high-precision molecular machinery ensures the equal partition of DNA between the two new-born cells [3] and determines the position of the dividing septum in the cell wall [4]. Particularly in *Staphylococcus* species, cells maintain spatial memory of previous division events via the location of cell-wall proteins [5], inducing a division pattern that alternates the formation of the new septum in geometrical planes orthogonal to previous cell division events (Figure 1) [6]. This mechanism has been extensively studied for *S. aureus* [7], and recent evidence suggests that it can be responsible for bacteriostatic effects delivered by surface topography [8].

The effect of nano-topography on biofilm formation has been investigated before for a range of materials displaying different structural features. Since the pioneering work of *Ivanova* et al. on bio-inspired surfaces [9], various antimicrobial topographies have been implemented, for instance, on silicon, titanium, aluminium oxide, and metal-alloy surfaces [8,9,10,11,12,13]. These topographies can vary in length-scales, geometrical properties, and mechanisms of action, and are effective against a wide range of microorganisms, including *Staphylococcus* spp., however, the theoretical understanding of key structure-activity relationships is still in progress [14].

On the other hand, surface attachment of *Staphylococcus* spp. is mediated by adhesins, whose mode of action has been only partially unravelled [15,16]. These cell-wall molecules can deliver different functions during surface colonisation, from the initial adhesion onto the surface to the recognition of host extracellular matrix molecules, driving the formation of biofilms. The versatile nature of these adhesins enables the attachment of *Staphylococcus* spp. to a variety of biotic and abiotic surfaces, using multivalent interactions, e.g., hydrogen bonds, electrostatic, etc [17]. Under this scenario, we can expect that the architecture of sessile micro-colonies of *Staphylococcus* spp. will emerge from the superimposition of these strong, localised cell-surface interactions and the topographically-constrained cell growth mechanism discussed above.

In this work we propose a topological model, supported by experimental evidence, to explain the geometric constraints that apply to the cell division of *Staphylococcus* species on surfaces with topographic cavities smaller than the dimensions of the single cells. Our model describes how the topological memory characteristic of the cell division mechanism of *Staphylococcus* spp. is responsible for bacteriostatic effects delivered by local surface topography. The model also allows quantification of the effect of surface topography on the architecture and surface attachment of the micro-colonies, showing that the number of surface anchoring points at the initial stages of biofilm formation can be drastically reduced by irregular topographies, such as those observed in vertically aligned silicon nanowire arrays.

## 2. Results and Discussion

### 2.1. Staphylococcus spp. on Flat Si: Lateral Freedom to Grow Away from the Surface

Attachment to surfaces is a widespread survival mechanism in bacteria that leads to the formation of biofilms [18,19]. Mature biofilms have been widely investigated and a range of defence mechanisms against environmental stressors has been described, mainly delivered by changes in cell phenotype and accumulation of extracellular substances [20]. Here, we focus on the early stages of biofilm formation, when the very first cells start to adhere and proliferate on a hard and chemically inert abiotic surface. During these first adhesion events, the symmetry of the cell surroundings is broken by the presence of the surface, but sessile cells are still able to grow and divide towards the surrounding environment, as demonstrated by scanning electron microscopy (SEM) imaging of surface-attached *staphylococcal* cells. (Figure 2).

Bacterial cells have developed complex mechanisms to recognise surfaces, and many of these sensing strategies have been only recently unravelled [21]. For motile bacteria, it is generally accepted that chemotaxis of chemical gradients, molecular interactions occurring at the cell-surface interface, and restriction of movement degrees of freedom upon attachment are instrumental in surface recognition and the subsequent change from planktonic to sessile phenotype [22,23]. However, for non-motile bacteria driven towards surfaces exclusively by Brownian motion, gravity, or other physicochemical interactions, these surface-sensing mechanisms are less understood [24].

The scenario complicates even further when we consider quasi-spherical non-motile cells, such as *Staphylococcus* spp., where the fraction of cell envelope in direct contact with any flat inert surface is minimal (i.e., a solid sphere touches a flat plane on a single point). Additionally, cell growth media involve the use of relatively high concentrations of salts, leading to a reduction in the characteristic length for cell-surface interactions. According to the extended formulation for the Deryaguin–Landau–Verwey–Overbeek (DLVO) theory of colloids [25,26], long-range interactions between cells and surfaces are dominated by electrostatic forces with a characteristic action length (a.k.a. Debye–Hückel length) that depends on the ionic strength of the medium. Conventional growth media (e.g., Luria–Bertani medium) yield Debye–Hückel lengths ≈1 nm, which are orders of magnitude smaller than the typical dimensions of bacterial cells (≈1 µm). This mismatch in length-scales is further enhanced by the curvature of quasi-spherical *staphylococcal* cells. Considering a cell with a radius of curvature ≈1 µm and a Debye–Hückel length ≈1 nm, we can estimate that upon close cell-surface contact, only 0.1% of the cell envelope would be influenced by the electrostatic forces of the surface. Similarly, for the very early stages of cell-surface attachment, chemical gradients from the metabolic activity of the cells are not yet present due to the low coverage of cells at the surface and the fast time-scales for diffusion of molecules in aqueous media. How these almost unperturbed cells are able to recognise the surface and change their phenotype, with only a marginal fraction of the cell envelope perturbed by the surface, is still to be understood.

The peculiar cellular morphology and cell-division mechanism of *staphylococci* provide an excellent model system to investigate cell growth and biofilm formation at topographic surfaces. In the early stages of the biofilm, these quasi-spherical cells usually display a single attachment point to the surface. They have also a sophisticated growth mechanism, orchestrated by the positioning of proteins at the peptidoglycan cell wall, carrying the memory of previous cell-division events [27]. This complex supramolecular system ensures that cell growth always occurs at 90° with respect to the previous cell-division event, alternating the geometrical planes where cell growth and colony development are allowed.

This behaviour of *staphylococcal* cell division has specific implications when attachment and growth on flat and topographic surfaces is considered. When the parent cell is attached to a hard surface, the alternating growth mode of *staphylococci* leads to the formation of local architectures in which the orthogonal cell-division geometry can be preserved after each division cycle (Figure 2G,H). Therefore, the primary orientation of the first coloniser cell and its cell-surface anchoring point, essentially determine the subsequent direction of cell growth for the 1st generation daughter cells and the allowed orientations for subsequent growth steps, thus defining the local architecture of the micro-colony at a surface.

For flat surfaces, different cases can be envisioned—e.g., growth direction for the 1st generation daughter cells parallel or perpendicular to the surface, and in any intermediate oblique orientation with respect to the surface plane, as experimentally observed in SEM images of different *Staphylococcus* spp. on flat silicon (Figure 2). On these flat surfaces, the lateral freedom of growth in the out-of-surface direction is always present, allowing for any cell to be able to grow and divide at the surface, regardless of its relative orientation with respect to the surface plane. However, in the following sections, we demonstrate how the presence of local irregularities in the topography of the surface may limit the ability of *staphylococcal* cells to grow, depending critically on the relative orientation of the first coloniser cell with respect to the surface plane.

### 2.2. Staphylococcus spp. on SiNW Surfaces: The Effect of the Local Topography

The effect of surface topography on cell growth and micro-colony architecture of *Staphylococcus* spp. was investigated using nano-patterned surfaces, displaying arrays of vertically aligned silicon nano-wires (SiNWs) referred to as ‘SiNW surfaces’ in this work. Reliable fabrication protocols published previously [28] enabled extended surfaces (i.e., 100 mm diameter silicon substrates) to be prepared with homogeneously distributed SiNWs, preferentially aligned to the normal direction with respect to the surface plane. Stochastic bending at the tips of the SiNWs generated a surface with irregular cavities displaying a broad size distribution, dominated by 100 nm to 300 nm pores (Figure 3). After fabrication, the SiNW surfaces were mechanically stable (i.e., non-deformable by capillary forces) and provided a hard, yet topographically irregular substrate, displaying 50.8% (±0.1%) of solid surface at the interface plane, with local fluctuations of pore size at the microscopic level and a long-range homogeneity of the pore size distribution across the sample. These SiNWs possessed a SiO_2_ layer at the surface due to the oxidation of Si during the fabrication process, and displayed low contact angles (<f10°) [28], therefore, the surface chemistry of this material was similar to the flat Si substrates bearing a SiO_2_ coating used in the previous section.

The typical pore size displayed by the SiNW array surfaces was smaller than the size of the *staphylococcal* cells, however the quasi-spherical shape of the cells meant that each cell essentially interacted with one single cavity at a time. The SiNW topographic surfaces did not show any intrinsic bactericidal effect. The cells were able to colonise and proliferate on the SiNW surfaces, showing similar coverages and % of viability compared to flat Si surfaces (Figure 4 and Table 1). Therefore, this type of surface provides a good model to investigate the separate effect of local topography on live, proliferating cells in the sessile state, attached to a chemically inert substrate. It is important to note that the data presented in Figure 4 were taken with a confocal microscope after 24 h of incubation, with the focal plane at the interface between the surfaces and the biofilm, to probe the potential contact-killing effect. These data are complementary to the high-resolution SEM images taken at shorter incubation time points, which capture early-stage biofilm growth and 3D morphologies of the micro-colonies.

A simple geometric model allows an evaluation of the constraints that govern spatially defined cell division events on hard surfaces (Figure 5). For cell growth on flat surfaces, the flat topography does not directly interfere with the cell division process, since lateral growth is allowed in any direction, both along and away from the hard surface, as schematised in Figure 5A. However, the presence of topographic cavities, such as those of the SiNW surface, changes this dramatically. The angular range in which cell growth and division are allowed is now limited by the boundaries of the local topography, determined by the maximum inscribed circle within each topographic cavity, represented in red within Figure 5B–D. Any attempt to grow below a certain critical angle defined by this cavity’s boundary will be hindered by the hard topography, creating a growth-blocking area, a_blocked_, on the cell envelope, as shown in red in Figure 5D. This analysis also defines the fraction of cell area where cell division is allowed, as shown in blue in Figure 5C,D (further geometrical considerations are presented in Appendix A). In principle, this growth-inhibition hindrance can go all the way from 0% on the perfectly flat surface (fraction of free area f_growth_ ≈ 1) up to 100% inhibition when the cell is completely trapped within the surface cavity (fraction of free area f_growth_ ≈ 0). Figure 5C depicts how the growth-inhibition hindrance increases as pore size increases.

If the direction of cell growth, which is predetermined by the intrinsic cell-memory mechanism of *Staphylococcus* spp. division, falls below the critical angle for a particular topographic cavity, cell division will be prevented, i.e., that cell will experience a bacteriostatic effect induced by the hard boundaries of the topography. The overall extent of growth inhibition on a particular surface will be determined by the size distribution of the cavities across this surface and by the typical size of the bacterial cells. Assuming a cell diameter of ≈1 µm, we calculated the weighted distribution for the fraction growth-allowed cellular area (f_growth_) over the SiNW surface, obtaining values between 0.80 and 0.99, with the average value at 0.95. These results suggest that, at the local level, the cavities generated by our SiNWs can induce local bacteriostatic effects of up to 20% growth inhibition when the local f_growth_ = 0.8, with an average of nearly 5% inhibition for the mean f_growth_ = 0.95 across the surface. This inhibition effect is relatively small, in agreement with the high cell coverage observed experimentally over the SiNW surfaces. The model proposed in Figure 5 is compatible with the recent work published by Jenkins et al. [8] showing bacteriostatic effects on *S. aureus* cells attached to TiO_2_ nano-pillar surfaces with surface structures that can completely accommodate the cells, a scenario for which our analysis would predict f_growth_ ≈ 0 (i.e., complete growth inhibition).

A direct implication of the local constraints induced by the topography of the SiNW surface is the critical angle for allowed cell growth and division. Once a primary colonising cell interacts with a topographic cavity, the growth of a 1st generation daughter cell is limited to angular values above that critical angle, determined by the radius of the cavity and the size of the cell. Therefore, only a fraction of the cell surface (f_growth_) can lead to cell division and colony growth (Figure 5D and Figure 6A). This local effect around the first surface anchoring point pushes micro-colonies to grow preferentially outside the surface plane on SiNWs, with daughter cells forced to propagate at a higher height, Δz, with respect to the surface plane. For the flat Si surface, a minimum Δz_min_ ≈ 0 is expected, since the first generation can grow parallel to the surface plane without any topographic hindrance. However, on SiNW topographies, Δz_min_ ranges between 200 nm and 600 nm, depending on the specific size of each pore (see Appendix A for details on the geometric formulations). The average Δz for a single cell can be determined as a function of the size of the cavity, using the pore size distribution shown in Figure 3. Following this approach, the probability density function of Δz offset values for SiNW topographies results in a Δz_average_ of between 700 nm and 950 nm, with the mean offset located at Δz_average_ = 750 nm (Figure 6B). These offset values are comparable to the characteristic size of *staphylococcal* cells, and would account for the preferential out-of-plane growth for the micro-colonies on SiNW surfaces (Figure 6C).

### 2.3. Local Propagation of Topological Information across Staphylococcal Micro-Colonies

In the previous section we demonstrated that the local effect of the SiNW topography constrains the 1st generation daughter cells to grow at a Δz offset from the surface plane, which averages between 700 nm and 950 nm. This strong effect on the local geometry of the colony around a primary anchoring point can affect the entire architecture of the colony. For example, any morphology with two contiguous cells attached to the surface will be topographically hindered on SiNW surfaces. In this section, we explore how this local topographic constraint may also influence the average number of anchoring points across the entire micro-colony.

As a first approximation, the architecture of *staphylococcal* micro-colonies can be represented using a mathematical formulation from graph theory. This approach considers each cell in the micro-colony as a graph-node connected to some other cells (nodes) by adjacency “bonds” and forming undirected mathematical graphs. The architecture of an ensemble of “n” cells can be, therefore, described by a symmetric adjacency matrix of dimensions (n × n):(1)$$C_{i}^{n}=\left[\begin{array}{ccc} a_{1,1} & \cdots & a_{n,1}\\ \vdots & \ddots & \vdots \\ a_{1,n} & \cdots & a_{n,n} \end{array}\right]$$
where each element of the matrix a_x,y_ will take binary values, i.e., a_x,y_ = 1 when cell “x” and cell “y” are connected, and a_x,y_ = 0 otherwise; while for every element of the matrix a_x,y_ = a_y,x_ (i.e., symmetry selection constraint). The diagonal elements of this adjacency matrix, which in graph theory are usually used to described mono-nodal loops, can in our case be exploited to denote surface attachment, namely, a_x,x_ = 1 if the cell “x” is attached to the surface, and a_x,x_ = 0 otherwise.

The dimension of the adjacency matrix $n=\dim(C_{i}^{n})$ corresponds to the total number of cells in the ensemble, while the modulus of the diagonal vector
(2)$$\overset{\leftarrow }{D_{i,n}}=\left|\overset{\leftarrow }{\mathrm{diag}}(C_{i}^{n})\right|$$
accounts for the number of contact points between the cells and the surface (i.e., the number of anchoring points of the micro-colony).

Under this approximation, and assuming that all allowed configurations are degenerated (i.e., having the same probability of occurrence), the average number of anchoring points for the ensemble of n cells 〈A〉_n_ on a flat surface can be calculated by the expression:(3)$${\langle A\rangle }_{n}=\frac{1}{M_{n}}\overset{M_{n}}{\underset{i=1}{\sum }}\left|\overset{\leftarrow }{D_{i,n}}\right|$$
where M_n_ is the total number of adjacency matrixes with dimension n, representing all allowed configurations for an ensemble of n cells. It is important to note that Equation (3) only considers colony architectures compatible with the geometrical constraints of the *Staphylococcus* growth-mode, discussed in Section 2.1, and shows alternating division events in orthogonal directions of the space. This geometric rule imposes a maximum of six adjacent daughter cells for all parent cells not directly connected to the surface, with a further reduction to a maximum of five adjacent daughter cells for parent cells attached to the flat surface. Applying these selection rules over each n^th^ order family of combinatorial adjacency matrixes, we can compute the value of 〈A〉_n_ as a function of n, using Equation (3). The results of these calculations are presented in Figure 7C.

Similarly, the average number of anchoring points for the SiNW surface can be calculated. In this case, due to the local effect of topography and the Δz offset imposed on first generation daughter cells originating from surface-borne parent cells, further topological restrictions must be considered. On the SiNW surface, configurations with two adjacent cells in the same plane of the surface are hindered, and therefore, any architecture and graph connectivity matrix showing at least two adjacent cells attached to the surface must be excluded from the calculations in Equation (3). At this point it becomes apparent that, within this first approximation, we are assuming that the information of the topographic constrains at a specific anchoring point is propagated between directly adjacent cells, but this local information is largely lost for higher order adjacency indexes. The rationale behind this assumption relies on the fact that the second generation daughter cells, non-adjacent to a surface-attached parent, can grow effectively in any direction within the plane orthogonal to the previous division event, and therefore the information of the topographic constraint is preserved only locally.

The simple topological selection rule discussed above, dictated by the effect of the topography on the cell division process, leads to a considerable reduction in the number of anchoring points for micro-colonies adhering to the SiNW surface. The average number of anchoring points on SiNWs diverges substantially from the value on flat surfaces as the number of cells increases (Figure 7C), and extrapolating for sufficiently large numbers of cells, we obtain:(4)$$\underset{n\to \infty }{\lim}\left(\frac{{\langle A\rangle }_{n}^{SiNW}}{{\langle A\rangle }_{n}^{flat}}\right)=0.4583$$

From these results we can infer that the average interactions between the micro-colonies and the surface are weaker on the irregular topography of SiNWs compared to flat Si, due to a reduced fraction of cells anchored to the surface.

The following section shows that this weaker attachment can be theoretically estimated for the entire population in terms of the Helmholtz free energy for sessile colonies ($\bar{{\langle \Delta F\rangle }_{n}}$), obtained by the balance between the entropy change for floating colonies to adhere to the surface ($\bar{{\langle \Delta S\rangle }_{n}}$) and the cell-surface interactions ($\bar{{\langle \Delta U\rangle }_{n}}$):(5)$$\bar{{\langle \Delta F\rangle }_{n}}=\bar{{\langle \Delta U\rangle }_{n}}-T\cdot \bar{{\langle \Delta S\rangle }_{n}}$$

In this expression, T is the temperature; the mathematical operator ${\langle X\rangle }_{n}$ denotes the average across the different micro-colony architectures of size “n”, as used above, while the operator $\bar{X}$ denotes the average across ensembles of different sizes within the entire population of micro-colonies. This expression can be further developed by introducing the average energy for one cell-surface interaction (Δε) and the entropy change per cell within the micro-colony (Δs), leading to:(6)$$\bar{{\langle \Delta F\rangle }_{n}}=\Delta \varepsilon \cdot \bar{{\langle A\rangle }_{n}}-T\cdot \Delta s\cdot \bar{n}$$

Under equilibrium conditions, at constant temperature and volume, $\bar{{\langle \Delta F\rangle }_{n}}=0$, and we can obtain a direct relationship between the average colony size ($\bar{n}$) and the average number of anchoring points ($\bar{{\langle A\rangle }_{n}}$) as follows:(7)$$\bar{n}=\frac{\Delta \varepsilon }{T\cdot \Delta s}\cdot \bar{{\langle A\rangle }_{n}}$$

Combining this expression with Equation (4), we can predict a reduction in the average size of the micro-colonies due to the topographic effect on SiNW surfaces:(8)$$\frac{{\bar{n}}^{SiNW}}{{\bar{n}}^{flat}}=\frac{{\bar{{\langle A\rangle }_{n}}}^{SiNW}}{{\bar{{\langle \Delta A\rangle }_{n}}}^{flat}}\approx 0.4583$$

This theoretical prediction is consistent with experimental results obtained from SEM data, discussed in the following section.

### 2.4. Effect of SiNW Topography on the Morphology of Staphylococcal Colonies

The key morphological descriptors for *staphylococcal* micro-colonies on flat Si and SiNW surfaces were obtained from low magnification SEM images, using a deep learning segmentation protocol (details on Section 3). This method allowed us to identify individual cells within each single colony and to calculate, directly from the SEM images (Figure 8A), three characteristic parameters for the micro-colonies, namely: the number of cells within the colony, the characteristic size of the colony (i.e., Feret’s diameter), and the colony’s circularity.

Comparing the populations of colonies attached to the two types of surfaces investigated here, we observe a 2-fold reduction on the average number of cells per colony, going from flat Si to SiNW surfaces, induced solely by the irregular topography of SiNWs (Figure 8B and Appendix A). This change in colony size, observed experimentally, can be rationalised in terms of the theoretical results discussed in the previous section, describing a reduction in the number of colony-surface anchoring points in the presence of the SiNW topography. This effect may be responsible for a preferential detachment of more fragile architectures within bigger colonies that cannot withstand mechanical action by Brownian motion and sheer forces from the surrounding medium, leading to an overall reduction in the average colony size. The reduction in colony size predicted theoretically by our model (≈0.46) is in good agreement with the experimental result (≈0.57) obtained by SEM imaging, suggesting that in a topologically-constrained system, such as sessile *staphylococcal* cells at the initial stages of biofilm formation, the local topographic information has a large determining effect on the fate of the system, dominating over other relevant factors (e.g., specific cell-surface adhesion factors, physical forces, etc.). Interestingly, the irregular topography of SiNWs also has some effect on the shape of attached micro-colonies (Figure 8C,D), inducing smaller Feret’s diameters and higher circularities, which correspond to less spread-out micro-colony shapes. These results are in agreement with the preferential out-of-plane growth-mode predicted by our topological model, suggesting that the local effect of the topography at the single cell-surface anchoring point also has an impact at the entire population level. Further generalisation of the result obtained here for *S. aureus* and *S. epidermidis* will require the incorporation into the model of other variables related to physical interactions and biological factors that are beyond the scope of the present work. Yet, the strong predictive power and the simple phenomenological interpretation of our model justify the relatively simple assumptions at its foundations.

## 3. Materials and Methods

### 3.1. Fabrication of SiNW Surfaces

SiNW surfaces were fabricated as reported before [28], starting from flat Si substrates (p-type boron-doped, (100)-orientation, resistivity 5–10 Ω cm, 100 mm diameter, thickness 525 µm, single side polished; Okmetic, Vantaa, Finland). Prior to further processing, the substrates were cleaned by immersion in fuming 100% nitric acid (HNO3, UN2031; OM Group, Cleveland, OH, USA) for 10 min and in boiling 69% nitric acid (HNO3, 51153574; BASF, Ludwigshafen, Germany) for 15 min, followed by rinsing in deionised water (DI water) and spin-drying. On silicon substrates, a regular pattern of 10 mm × 10 mm squares was defined with UV-lithography, and each square comprised a centred area of 8 mm × 8 mm in which SiNWs were subsequently obtained. After development, the patterned resist was post-baked for at least 10 min at 120 °C in ambient air. SiNWs were formed in the lithographically patterned areas following a two-step metal-assisted chemical etching (MACE) process [29,30]. A 50% aqueous hydrofluoric acid (HF, 51151083; BASF, Ludwigshafen, Germany) was diluted 5 times in DI water to which silver nitrate (AgNO_3_; Sigma-Aldrich, Zwijndrecht, The Netherlands) was added at a concentration of 5 mM. In order to deposit Ag nanoparticles (AgNPs) on the exposed silicon areas, the patterned substrates were immersed in this solution and were kept in the dark for 1 min. Subsequently, the substrates were directly loaded in a solution of DI water, 50% aqueous hydrofluoric acid (HF, UN7190; BASF, Ludwigshafen, Germany) and hydrogen peroxide (H_2_O_2_, 511316830; BASF, Ludwigshafen, Germany) (volumetric ratio DI:HF:H_2_O_2_ = 77.5:20:2.5), and etched for 20 min in the dark (etch rate ca. 0.6 µm min^−1^). After this, the substrates were rinsed with DI water. The AgNPs were removed by immersing the substrates in 69% nitric acid (HNO3, 51153574; BASF, Ludwigshafen, Germany) at room temperature for 65 h, which was followed by a rinsing step with DI water. To ensure complete removal of traces of photoresist, the substrates were cleaned in Piranha-solution (a 3:1 volumetric mixture of sulphuric acid (H_2_SO_4_, UN1830; BASF, Ludwigshafen, Germany) and H_2_O_2_; temperature 95 °C, cleaning time 15 min), after which they were rinsed with DI water and dried with nitrogen. Finally, individual samples of 10 mm × 10 mm were cut using a dicing machine (Disco DAD-321; Disco Hi-Tech Europe GmbH, Munich, Germany).

### 3.2. Imaging of Sessile Bacteria on Flat Si and SiNW Surfaces

*S. aureus* (DSM 346, DSMZ-German Collection of Microorganisms and Cell Cultures GmbH, Braunschweig, Germany) and *S. epidermidis* (ATCC 12228, Manassas, VA, USA) were transferred from frozen stock to a fresh agar plate and incubated overnight at 37 °C. Three colonies of each bacterial species were taken from the agar plate with a sterile plastic loop, transferred to fresh nutrient broth (NB) medium (Thermofisher, Waltham, MS, USA) and grown overnight in a shaking incubator (200 rpm, at 37 °C). SiNW surfaces and flat Si wafers underwent sterilisation under UV light for 20 min and were then placed in a sterile 24 well plate. Subsequently, 1 mL of 10^5^ CFUs/mL bacterial suspension in NB was added to each of the wells and the samples were incubated for 8 h or 24 h at 37 °C. Viability and imaging assays using bacteria were performed in triplicate.

### 3.3. Viability of Sessile Bacteria

The viability of bacteria adhering to surfaces was studied using confocal laser scanning microscopy (CLSM, Zeiss, Oberkochen, Germany) with protocols optimised before [31]. Briefly, following the incubation period of 24 h, the samples were gently washed 3 times with sterile 0.85% (*w*/*w*) NaCl solution and stained with Live/Dead BacLight bacterial viability kit (Molecular Probes, L7012, Eugene, OR, USA). In this test, each sample was incubated in a 24-well plate for 15 min in the dark at room temperature in 1 mL of a sterile 0.85% solution containing a mixture of SYTO 9 (Molecular Probes, Eugene, OR, USA) and propidium iodide (PI) (Molecular Probes, Eugene, OR, USA). After staining, the samples were immediately imaged using a Zeiss LSM 880 upright Multiphoton microscope (CLSM, Zeiss, Oberkochen, Germany). Collected confocal fluorescence images were processed using Fiji image analysis software.

### 3.4. Scanning Electron Microscopy (SEM) Imaging

For SEM analysis, samples were processed following previously reported protocols [32]. After 8 h of incubation, the samples were fixed overnight at 4 °C in a 4% paraformaldehyde (TAAB, Berks, UK) and 2.5% glutaraldehyde (TAAB, Berks, UK) in 0.1 M phosphate buffer (ph = 7) (Thermofisher, Waltham, MS, USA). Sample fixation was progressed by three sequential steps at room temperature involving aqueous solutions of 2% osmium tetroxide (TAAB, Berks, UK) for 1 h, 1% tannic acid (TAAB, Berks, UK) for 30 min and 2% osmium tetroxide for 1 h, followed by overnight incubation in 1% uranyl acetate solution in water at 4 °C. Abundant rinsing with DI water was performed between each fixation step. The samples were then rinsed with DI water and progressively dehydrated using increasing ethanol concentrations (i.e., 30%, 50%, 70%, 90% and 100%), CO_2_-critical-point dried (Quorum Technologies K850, Sacramento, CA, USA) and subsequently sputter coated with 10 nm of Au/Pd (Quorum Technologies Q150T, Sacramento, CA, USA) for SEM imaging at 10 kV using a JEOL7001F FE-SEM system (Jeol LTD, Akishima, Japan).

### 3.5. Quantification of Pore Size of SiNW Surfaces

SEM images of SiNWs were processed and quantified using a homemade macro for Fiji image analysis software [33]. The complete pipeline is publicly available in https://github.com/ioritzsb/Cocci_on_SiNW (accessed on 15 January 2022). Prior to pore size quantification, SEM images were binarised using ImageJ’s Default threshold algorithm, and denoised using a median filter. Then, the mask fraction representing the pores was filled with the largest inscribed circles [34] using a minimum circle diameter of 10 pixels, and the corresponding diameters were subsequently exported as a tab separated values file (.tsv) for data analysis.

### 3.6. Characterisation of Colony Morphology

A deep learning segmentation model for *S. aureus* SEM images on flat and SiNW surfaces [35] was trained from scratch for 100 epochs on 8 paired image patches (image dimensions: (200, 200), patch size: (176,176)) augmented by a factor of 4, (Bloice, Marcus D., Christof Stocker, and Andreas Holzinger, “Augmentor: an image augmentation library for machine learning.” arXiv preprint arXiv:1708.04680 (2017)) with a batch size of 2 and a mean absolute error loss function (Appendix A), using the StarDist 2D [36], ZeroCostDL4Mic notebook (v 1.12.3) [37]. Key parameters and libraries used for the StarDist 2D model generation are displayed in Appendix A, respectively. Then, the StarDist 2D model was applied on unseen SEM images to obtain labelled segmentation masks, which were further processed using a GPU accelerated [38] homemade script for Fiji image analysis software (see https://github.com/ioritzsb/Cocci_on_SiNW. accessed on 15 January 2022), to obtain characteristic descriptors of S. aureus colony morphology on the different surface types.

### 3.7. Data Analysis

Weighted probability density functions of the pore size (r_pore_), fraction of growth-allowed cell area (f_growth_), average vertical offset (Δz), and average critical angle θ for the SiNW surfaces were calculated as described in Table 2 and represented by calculating a Gaussian kernel-density estimate using seaborn [39] statistical data visualisation package from Python.

Characteristic descriptors of *S. aureus* colony morphology on flat (*N* = 1341) and SiNW (*N* = 1230) surfaces—i.e., weighted number of cells per colony, colony circularity and Feret’s diameter-—were tested for normality using Kolmogorov–Smirnov and Shapiro–Wilk test (*p* < 0.0001 in both tests for all the distributions) and subsequently compared using non-parametric Mann–Whitney U test.

### 3.8. Computations of Graph Adjacency Matrixes

Adjacency matrices for micro-colony architectures were generated by a combinatorial algorithm implemented in Python 3.6 and selecting only symmetric matrices. We subsequently applied to this combinatorial ensemble of matrices the geometric constrains of *Staphylococcus* cell growth and division mechanism, requiring alternating division events in orthogonal directions of the space, which imposes a maximum of 6 adjacent daughter cells for all parent cells non-directly connected to a surface, and a further reduction to a maximum of 5 adjacent daughter cells for parent cells that are surface attached. This rule applies to both flat Si and SiNW surfaces, and can be enforced on the entire family of adjacency matrixes {a_i,j_} as the following constrain:(9)$$\forall \;i\;\left(\underset{j\neq i}{\sum }a_{i,j}\right)<\left(6-a_{i,i}\right)$$

On SiNW surfaces, there is an additional constrain, imposed by the effect of local topography on *Staphylococcus* cell growth and division, and forbidding configuration with adjacent cells on the same plane of the surface, which leads to the constrain:(10)$$\left(\underset{i}{\sum }\underset{j\neq i}{\sum }a_{i,j}\cdot a_{i,i}\cdot a_{j,j}\right)=0$$

These constrains were then applied to the ensemble of adjacency matrixes to obtain the sub-family of graphs allowed on SiNW surfaces. The average number of anchoring points as a function of the number of cells (n) was then calculated independently over the sub-families of adjacency matrix allowed on flat Si and SiNW surfaces, using Equation (3). The computed values were then numerically fitted using simple linear regression in Graphpad Prism 9.0.2 (Details of the best numerical fittings are presented in Appendix A).

## 4. Conclusions

We developed a topological model supported by direct experimental evidence and able to explain micro-colony architecture and cell division of *Staphylococcus* species on vertically aligned SiNW arrays, with topographic features smaller than the dimensions of the single cells. We demonstrated experimentally that these surfaces do not deliver a direct contact-killing effect, but still show a moderated bacteriostatic action due to the surface topography and a drastic effect on the local architecture of the colonies. Our model explains these findings based on the characteristic topological memory of cell division mechanism in *Staphylococcus* spp., responsible for a discrete bacteriostatic effect delivered by the surface topography. Our topological model also allows the direct quantification of the effect of surface topography on the number of anchoring points and on the average size of the colonies that appeared significantly reduced by irregular topographies even at the initial stages of biofilm formation. These results may pave the way for new strategies to prevent biofilms via surface engineering at the nanoscale.

## Figures and Tables

**Figure 1 nanomaterials-12-00683-f001:**
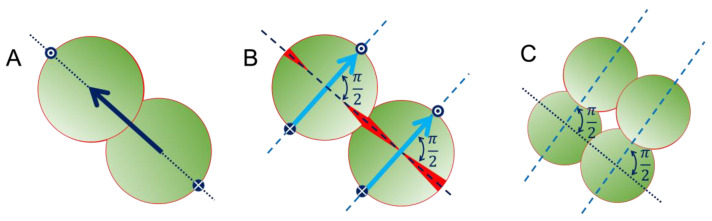
Schematic representation of the spatial orientation of subsequent cell-division cycles for *Staphylococcus* spp., showing that the new septa (in red) are formed in the line containing the direction of the previous division, indicated by a dark blue arrow in (**A**), and determining the direction of the next division cycle in the orthogonal plane (**B**,**C**). This process is regulated by the positioning of specific proteins at the cell envelope, represented by blue dots.

**Figure 2 nanomaterials-12-00683-f002:**
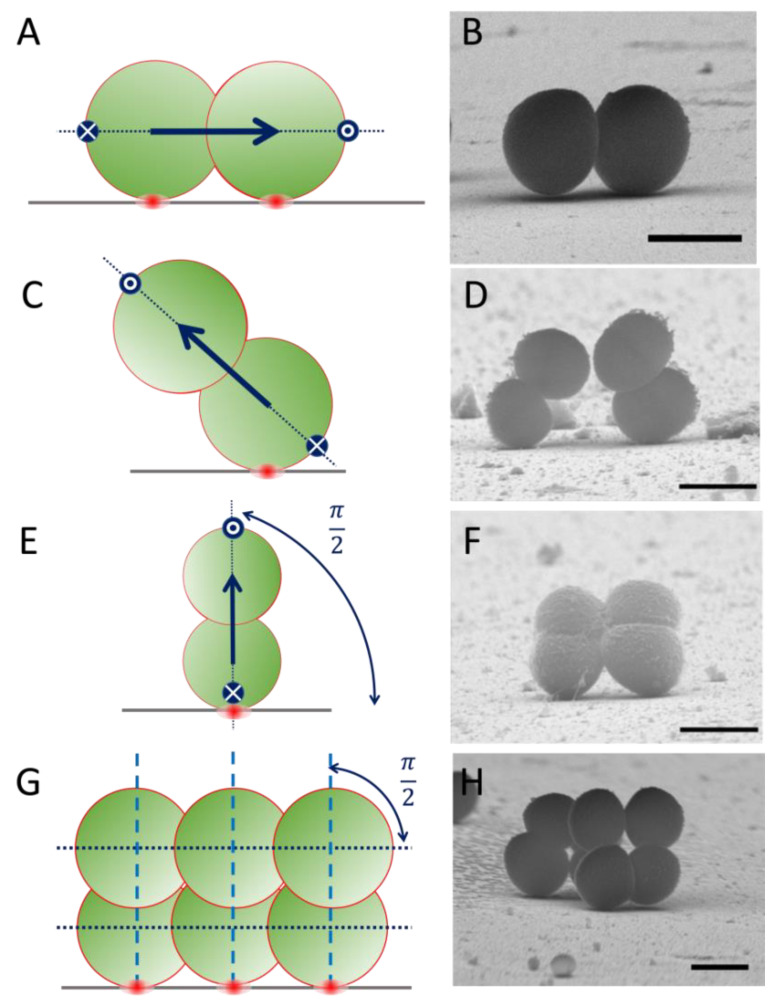
Cell-division of *Staphylococcus* spp. on flat Si surfaces. Left column: schematic representation of cell division; Right column: representative Scanning Electron Microscopy (SEM) images of *staphylococcal* cells on flat Si surfaces. (**A**–**F**) After initial attachment, the symmetry of the cell environment is broken but the cell is able to progress with cell growth and division towards the surrounding environment. The orientation of initial cell-surface attachment (red dot) can determine the direction (blue arrow) of subsequent cell division events on *Staphylococcus* spp. On flat surfaces, cell growth may occur at any angle relative to the surface plane. (**G**,**H**) Due to the topological memory of the cell division mechanism of *Staphylococcus* spp., the 1st generation daughter cells grow in an orthogonal direction with respect to the initial division event. Scale bars 1 µm. Note: ((**B**) *S. epidermidis*; (**D**,**F**,**H**)) *S. aureus*.

**Figure 3 nanomaterials-12-00683-f003:**
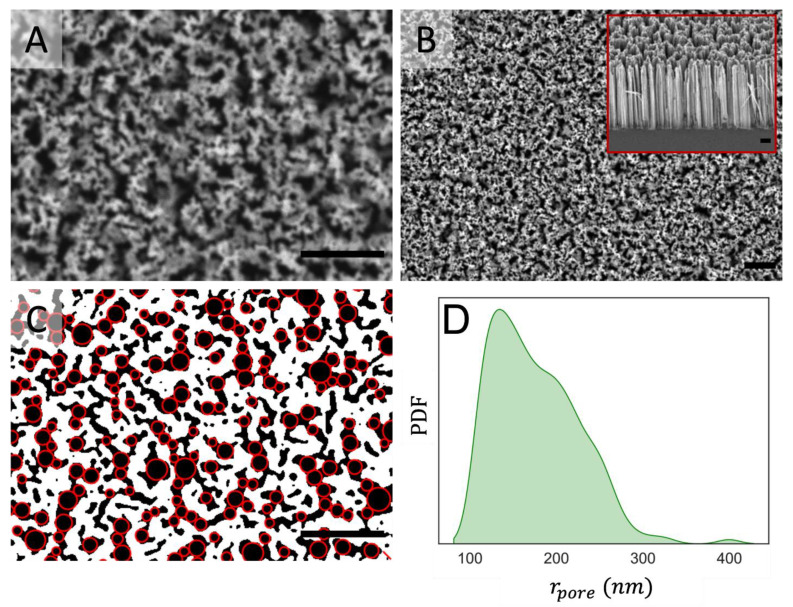
Scanning electron microscopy (SEM) data for vertically aligned silicon nanowire arrays: (**A**,**B**) Top-view SEM images at different magnifications, inset in B displays SEM side view of the SiNW surface; (**C**) Representative segmentation mask generated from SEM images, showing silicon nano-wire arrays (white), pores (black) and inscribed circumferences (red) used to calculate the pore size; (**D**) Probability density function of the pore radius for SiNW surfaces weighted by the pore area. Scale bars are 2 µm.

**Figure 4 nanomaterials-12-00683-f004:**
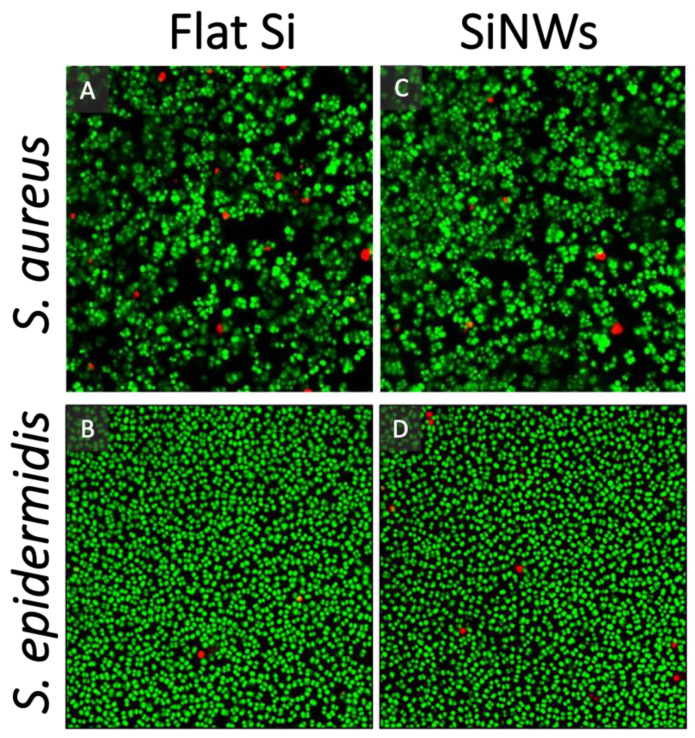
Representative Live/Dead Confocal Scanning Laser Microscopy (CSLM) images for *Staphylococcus* spp. stained with Syto9 (green, live) and propidium iodide (red, dead) cultured for 24 h over: (**A**,**B**) Flat silicon; and (**C**,**D**) SiNW surfaces. Field of view is 71.8 µm × 71.8 µm. The confocal images were taken at the interface between the surfaces and the biofilm, showing only the first layer of cells in contact with the surface.

**Figure 5 nanomaterials-12-00683-f005:**
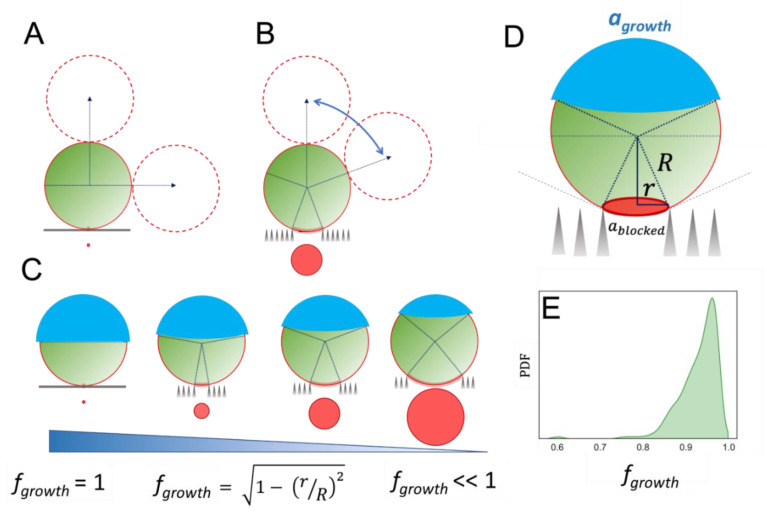
Effect of local topography on cell growth and division: (**A**) On a flat surface, any orientation angle is allowed for subsequent cell division. (**B**) In the presence of a topographic cavity, the allowed angles for subsequent cell division are restricted by the boundary of the cell that lies within the topographic cavity, as shown in red. (**C**,**D**) The topographic restriction of the cavity generates a hindered zone on the cell wall, determined by the maximum red circle inscribed in the cavity (**D**). This also defines the fraction of cell area *f_growth_* where cell division is allowed, depicted in blue (**C**,**D**). In the equation displayed in (**C**), *r* stands for the pore radius and *R* for the radius of *Staphylococcus* spp. cells as illustrated in (**D**). (**E**) Probability density function for f_growth_ on SiNW surfaces weighted by the pore area.

**Figure 6 nanomaterials-12-00683-f006:**
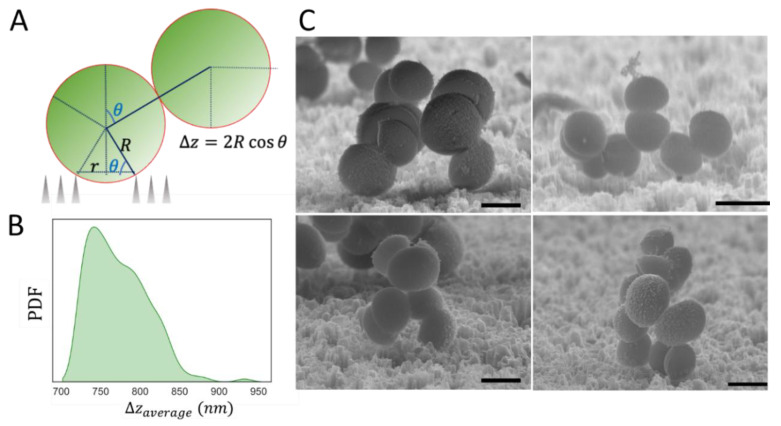
Effect of local topography on the critical angle for allowed cell growth: (**A**) The radius of the cavity (r) and the size of the cell (R) determine the critical angle below which cell division is hindered, imposing a Δz offset for the daughter cells. (**B**) Probability density function of the average Δz offset values over the SiNW surface for spherical cells of diameter ≈ 1µm. (**C**) Representative SEM images of *S. aureus* micro-colonies on SiNW surfaces. Scale bars are 1 µm. Additional SEM data are presented in Appendix A.

**Figure 7 nanomaterials-12-00683-f007:**
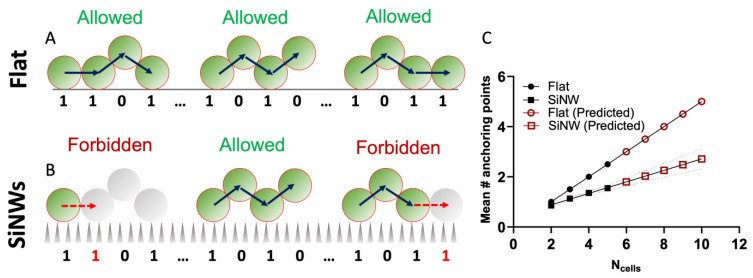
Effect of local topography on the average number of anchoring points for an ensemble of spherical cells: (**A**,**B**) Schematic representation of the topological selection rule from the model developed here; (**C**) Theoretical average number of anchoring points as a function of total number of cells for flat Si and SiNW surfaces. Red points are extrapolated from the fitting equations presented in Appendix A.

**Figure 8 nanomaterials-12-00683-f008:**
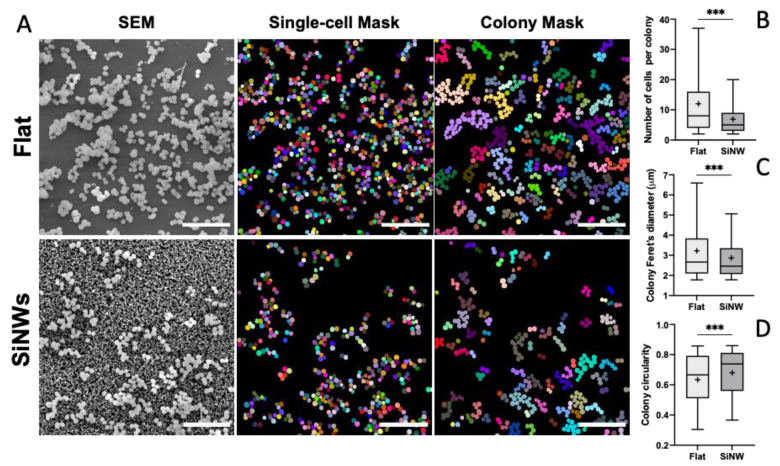
Morphological characterisation of *S. aureus* micro-colonies on flat Si and SiNW surfaces: (**A**) Top view SEM images, individual cell mask and single colony mask for image segmentation (Scale bars 10µm); (**B**) Weighted number of cells per colony; (**C**) Feret’s diameter for the colonies (in µm); (**D**) Colony circularities on flat Si and SiNW surfaces with error bars showing the standard error of the mean. *** indicates high statistically significant difference between groups (*p* < 0.001). Boxplots represent the median (horizontal line), mean (+), interquartile range (box), bottom 5% value (lower whisker) and 95% top value (upper whisker).

**Table 1 nanomaterials-12-00683-t001:** Fraction of live cells on flat Si and SiNW surfaces ^1^.

*Staphylococcus* spp.	% Live Cells on Flat Si Surface	% Live Cells on SiNW Surface
*S. aureus*	98.1% ± 3.5%	97.7% ± 9.3%
*S. epidermidis*	98.9% ± 4.9%	98.5% ± 7.9%

^1^ Data calculated from Laser Scanning Confocal Microscopy images using Live/Dead staining. Further details in Section 3.

**Table 2 nanomaterials-12-00683-t002:** Key topographic parameters investigated in this work (detailed description of geometric considerations is presented in Appendix A).

Parameter	Calculation
$r_{pore}$	Determined from SEM images (see Appendix A)
$f_{free}$	$f_{free}=\sqrt{1-\frac{{\left(r_{pore}\right)}^{2}}{{\left(R_{cell}\right)}^{2}}}$
$\Delta z_{average}$	$\Delta z_{average}=\frac{2\cdot R_{cell}\cdot \sin\left(\theta_{max}\right)}{\theta_{max}}$
θmax	$\theta_{max}=arcos\left(\frac{r_{pore}}{R_{cell}}\right)$

## Data Availability

The data that support the findings of this study and all custom codes are available from the corresponding author upon reasonable request.

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
