# Peer review of "Effect of Local Topography on Cell Division of Staphylococcus spp."

_nanomaterials, 2022, doi:10.3390/nano12040683_

Round 1

Reviewer 1 Report

The manuscript presented by Bellido et al. provides a clear demonstration of the critical influence of the surface topography towards bacteria ability to adhere and colonize an implant surface.

The topic is of great interest as bacterial infections represent nowadays a major reason of implant failure as well as the raising antibiotics resistance displayed by many pathogens requires for the development of alterantive strategies to prevent infections.

Here the Authors provided a clear and realistic model to compare flat and nanostructured surfaces by correlating the possibility for bacteria growth and adhesion in function of the directions allowed by the surface itself.

The results are clear and logically related to the hypothesis, as well as they seem to be in line with the majority of the previous literature suggesting that particular topographies can prevent bacteria adhesion ar adaptation to the surface due to the cytoskeleton stress in contact with the surface irregularities.

Therefore, in my opinion this work can be accepted for publication after minor changes.

  1. I was expecting an oriented distribution of the nanowires on the surfaces and not a randomic one (as showed in figure 3C) to maximize the effect of the topography. So, even if the results clearly demonstrated the differences in comparison to the flat surface, the random geometry open for the discussion about how the different pores/arrays distribution and clusters influence (or not) bacteria adhesion. This should be considered and for future experiments I suggest to move to a more controlled topography to minimize the number of variables.
  2. The presence of the nanowires can affect not only the surface topograhy but also the physical/chemical properties such as surface energy and wettability that are responsible for antofouling properties too (see for example the recent work DOI:10.1016/j.mtbio.2021.100148). Those parameters should be evaluated (or at least considered) also for the model here proposed by the Authors.
  3. The references' list seem to be short, some other works can be cited maybe with reference towards materials suitable for implantation (titanium, ceramics etc).

Author Response

We thank the reviewer for the useful comments and reply all the queries in the revised manuscript and below:

1) Orientation of SiNWs: Our surfaces contain vertically aligned wires (see added image in Fig 2). However, due to local bending-fluctuations, the tips of the wires randomly touch each other, creating the irregular topography observed in Figure 3C. The relatively easy implementation and scale-up of the fabrication process provides a good platform for initial evaluation of topographic effects on cell growth and division (i.e. the system has vertically aligned wires, but the surface displays cavities of different sizes, allowing single colony-surface interaction-events to be proven by high-resolution imaging, while exploiting the range of cavity sizes within the same sample). As suggested by the Reviewer, for future work we will aim at the optimisation of the fabrication process to obtain mono-dispersed topographic cues, but at present it would require an experimental effort beyond the scope of this paper.

2) Surface properties of SiNWs: The surface properties of SiNWs arrays have been investigated before (see manuscript Ref 19). The wires display a thin layer of SiO2 at the surface that provides low contact angles (<10°), very similar to flat SiO2, allowing direct comparison between flat and topographic surfaces with the same surface chemistry. We have included additional statements in the revised manuscript to highlight these facts and to answer the Reviewer's comments.

3) Expanding bibliography: We have expanded the reference list and the introduction to account for the Reviewer's comment.

Reviewer 2 Report

In my opinion a very interesting article, well written, with appropriate methodology and results evaluation and discussion. Therefore, I have no major comments or suggestions to make.

However, I would like to see a more complete introduction. There is published work concerning the effect topography cues on the antimicrobial properties of several type of materials used in everyday applications (not the case of silicon) that should be addressed in the introduction.

Author Response

We thank the reviewer for the positive comments.

In the revised manuscript we have expanded the introduction and included references to a wider literature on the topographic effect for different materials, as requested by the Reviewer.

Reviewer 3 Report

In this manuscript, the authors answered the very interesting topic that local surface topography governs staphylococcal adhesion to abiotic surfaces. This manuscript is well structured and written and provided many descriptions to support their opinions on the microorganism's adhesion process. However, a few issues need clarification before the manuscript can be accepted for publication.

Figure 4 shows confocal microscopic images of Staphylococcus aureus and Staphylococcus epidermidis after 24 h growth on flat silicon and silicon nanowires arrays.

At first glance, it shows the typical microcolony structure (after 24 hours of growth, you can already observe a biofilm) for S. aureus and the unexpected arrangement of single cells ???? for S. epidermidis. Therefore, there are apparent differences between these two species of staphylococci. Other photos from SEM and different modeling results are labeled on the following manuscript figures as "Staphylococci sp.", without specifying species affiliation. So for what species were all these calculations and modeling carried out? How do the authors interpret these differences in the surface colonization of the two species of staphylococci? I am also asking for a precise specification of what staphylococci species is presented in the figures.

Secondly, the adhesion of staphylococci to the abiotic surfaces occurs in the presence of proteins. The expression "race to the surface" is well known and is mentioned when considering the early stages of bacterial adhesion. The authors should at least mention this phenomenon and argue that the presence of proteins will not affect their model (see the recent works, e.g., doi.org/10.3389/fmicb.2021.686793). What if adhesion occurs via MSCRAMMs molecules' attachment to extracellular matrix proteins? It is already known that there are subtle changes in the adhesive particles when in contact with nanoparticles (e.g., DOI: 10.1016 / j.micpath.2020.104239)

The authors should at least discuss these issues. Their model does not mention these observations, simplifying the phenomenon of adhesion. However, the title of the manuscript and the subsequent discussion of the rough surface bacteriostatic effect may lead to far-reaching conclusions regarding the modification of implant surfaces.

Hence, I believe that these issues should be briefly discussed, pointing to the limitations of the developed model.

Finally, please correct the title and then the entire text of the manuscript – change from Staphylococci sp. to the correct name: Staphylococcus spp.

Author Response

We thank the Reviewer for the constructive comments and reply to all the queries in the revised manuscript and specific responses below.

Bacterial species: We focused our experimental work on two representative Staphylococcus species, namely  S. aureus and S. epidermidis, that are commonly used as model microorganisms to investigate cell division and growth, surface attachment and biofilm formation. We have corrected all the figure captions to specify species affiliation.

We investigated the initial stages of the biofilm and recorded high-resolution images of cell-surface anchoring events that allowed us to conclude that both species behave similarly at the local level (see Supporting Information Figures S1.1 & S1.2). This result is consistent with the fact that surface attachment and the cell division mechanisms are well-preserved among Staphylococcus spp.

Adhesion mechanism: We have revised the manuscript introduction to include surface adhesion mechanisms for Staphylococcus spp., adding the literature references suggested by the reviewer. Our model aims to describe the effect of topography on cell division and colony growth, after surface adhesion is consolidated. Therefore, we cannot comment explicitly on  the initial adhesion phase. However, given that topographic effects are observed, one can assume that the presence of proteins does not destroy the underlying topographic effect.   

Typos: The manuscript has been proof read and typographical mistakes corrected, including the spelling of Staphylococcus spp.

Round 2

Reviewer 3 Report

I want to thank the authors for the changes to the manuscript text.
However, I still have a problem with Figure 4. We observe an utterly different biofilm morphology after 24 hours of growth of both species of bacteria. In my opinion, one of two situations could arise:
1. the biofilm morphology, i.e., the outcome of the proposed model, is different for S. aureus and S. epidermidis;
2. Figure 4 is mistakenly signed, and a similar biofilm morphology for these two species occurs on flat surfaces and is different from that on nano-patterned surfaces.
If Figure 4 is properly signed, the Authors should comment on these differences and how the proposed model considers this phenomenon.
If the Figure is erroneously signed - it should be corrected. But then it follows that scarce (no) biofilm is formed on nano-patterned surfaces.
The authors ignored this problem in their responses to the reviewer's questions.
In addition, in the caption for Figures 6 and 8, Staphylococcus aureus is in the full name, in other captions as S. aureus; it should be harmonized according to the editorial requirements.

Author Response

We thank the referee for the comments, which are addressed below:

The morphologies of the 24 hr biofilms, as observed by the confocal microscope focused at the interface between the biofilm and the surface, show differences between S. epidermidis and S. aureus. However, for the interpretation of the data presented in Fig4 in relation to the rest of the work, several considerations must be taken into account:

  1. the confocal images shown in figure 4 were taken after 24h of incubation, setting the focal plane at the interface between the surface and the biofilm to effectively probe the potential contact-killing effect of the topography. The scope of this experiment was simply to evaluate the biocidal effect of the surface, and no information on the 3D structure of the colonies or the attachment points at the surface can be directly inferred.
  2. To investigate (and model) the effect of surface topography on the local architecture of the microcolonies we needed to focus on the early stages of the biofilm, where the relatively low coverage of the surface allowed us to probe single cell-surface anchoring points with sufficient resolution by SEM. This relatively low coverage (needed for the main scope of our work) was not appropriate for accurate quantification of Live/Dead images. For this reason, we chose different time points for SEM and Live/Dead imaging, and direct comparison of these complementary data sets is not straightforward.
  3. Our SEM data of S. aureus and S. epidermidis capture the early stages of biofilm growth and show very similar morphologies for the microcolonies (additional images have been added to SI to support this finding) suggesting that our model is applicable in both cases.

Following these considerations, we have addressed the Reviewer's concerns by adding clarifications to the caption in Fig4, additional statements in the discussion, and additional images in SI to support our interpretation. In detail, the text added to the discussion of Fig4 is cited below:

It is important to note that the data presented in Figure 4 was taken with a confocal microscope after 24h of incubation, with the focal plane at the interface between the surfaces and the biofilm, to probe the potential contact-killing effect. This data is complementary to the high-resolution SEM images taken at shorter incubation time points, which capture early stage biofilm growth and the 3D morphologies of the micro-colonies.